# Antimicrobial Susceptibility Profile of Rare Anaerobic Bacteria

**DOI:** 10.3390/antibiotics12010063

**Published:** 2022-12-29

**Authors:** Lena Josephine Wolf, Catalina-Suzana Stingu

**Affiliations:** Institute for Medical Microbiology and Virology, University Hospital of Leipzig, 04103 Leipzig, Germany

**Keywords:** antimicrobial susceptibility, anaerobic bacteria, E-test, agar dilution method

## Abstract

Anaerobes play an important role in clinically relevant infections and resistance is increasing worldwide. We tested 120 rare anaerobic isolates belonging to 16 genera for antimicrobial resistance using the agar dilution method and compared those results to the time-saving E-test method. The susceptibility data for 12 antimicrobial substances (benzylpenicillin, ampicillin/sulbactam, piperacillin/tazobactam, imipenem, meropenem, cefoxitin, metronidazole, moxifloxacin, clindamycin, doxycycline, tigecycline, eravacycline) were collected. Susceptibility testing showed low resistance to β-lactam/β-lactamase inhibitor combinations and no resistance to carbapenems and tigecycline. We observed moderate to high rates of resistance to moxifloxacin and clindamycin which differed depending on the methodology used. The essential and categorical agreement was over 90% for ampicillin/sulbactam, meropenem, moxifloxacin, and tigecycline. For metronidazole and clindamycin, the essential agreement was below 90% but the categorical agreement was near or above 90%. Penicillin presented with the lowest categorical agreement of 86.7% and a very high very major error rate of 13.3%. The resistance rates reported in this study are concerning and show the importance of routine susceptibility testing. Further investigations are necessary to determine the reason for high error rates and how to improve susceptibility testing of fastidious anaerobes.

## 1. Introduction

Anaerobic bacteria make up the greater part of the human commensal microbiota. They can cause endogenous infections when the immune system is compromised or mucosal barriers are damaged [1]. Often, these infections are mixed, with multiple anaerobic and aerobic bacterial species present [2]. These infections can be life-threatening, so appropriate antimicrobial therapy is life-saving [3]. Because of the fastidious nature of some anaerobes, isolation and routine susceptibility testing can prove to be difficult [4]. Therefore it is important to provide current data on antimicrobial susceptibility patterns.

For many rare species tested in this study, very little current data on resistance is available. Additionally, due to recent taxonomy changes [5], it is becoming increasingly difficult to compare antimicrobial susceptibility to previously published studies. Nevertheless, it is clear that resistance among anaerobes is generally increasing worldwide [1,6,7], and susceptibility should be assessed locally because resistance rates vary depending on the region.

EUCAST (European Committee on Antimicrobial Susceptibility Testing) recommends determining the minimum inhibitory concentrations of anaerobic bacteria using the agar dilution method [8]. Although this method is very accurate, it is time-consuming and not suitable for routine testing. E-tests are much easier to use and regularly utilized in routine testing. Because the species that were tested in this study are rare, it is unclear if E-tests provide reliable results. This study aims to provide susceptibility data for rare anaerobes and to compare the gold standard agar dilution method to the E-test methodology.

## 2. Results

Overall, antimicrobial susceptibility testing with the agar dilution method showed no resistance to tigecycline, imipenem, and meropenem. The minimum inhibitory concentrations (MIC) determined by agar dilution method are listed in Table 1 and the MICs determined by E-tests are listed in Table 2. If not stated otherwise, the susceptibility profiles presented in this section are determined through agar dilution testing, as it is widely regarded as the gold standard method. For the antimicrobial substances with no EUCAST breakpoints (cefoxitin, doxycycline, and eravacycline) no susceptibility rates could be calculated. The low MICs of eravacycline suggest that all of the isolates tested were susceptible, although formal categorizing is not possible due to the lack of breakpoints.

*Prevotella* spp. including *P. salivae* (*n* = 5), *P. melaninogenica* (*n* = 4), *P. baroniae* (*n* = 3), *P. timonensis* (*n* = 3), *P. buccae* (*n* = 2), *P. denticola* (*n* = 2), *P. bergensis* (*n* = 1), *P. disiens* (*n* = 1), *P. loescheii* (*n* = 1), *P. nanceiensis* (*n* = 1), *P. nigrescens* (*n* = 1), and *P. orachea* (*n* = 1) were very susceptible to β-lactam/β-lactamase-inhibitor combinations. The benzylpenicillin resistance rate was 36% (9/25), and especially common in *P. timonensis* (3/3) and *P. melaninogenica* (3/4). Resistance to moxifloxacin was seen in 88% of isolates (22/25). Metronidazole and clindamycin resistances determined by agar dilution method were 12% and 24% respectively, while the E-test method showed no resistance to metronidazole and only 16% resistance to clindamycin.

*Capnocytophaga* spp. including *C. gingivalis* (*n* = 14), *C. granulosa* (*n* = 3), *C. orachea* (*n* = 3), *C. sputigena* (*n* = 1), and *C. ureolyticus* (*n* = 1) showed susceptibility to β-lactam/β-lactamase-inhibitor combinations. Metronidazole resistance was high at 82%. The resistance rate to clindamycin was very high in *C. gingivalis* (13/14), *C. sputigena* (1/1), and *C. ureolyticus* (1/1). *C. granulosa* and *C. orachea* isolates showed susceptibility to clindamycin. *C. ureolyticus* showed resistance to benzylpenicillin, piperacillin/tazobactam, moxifloxacin, and clindamycin.

*Fusobacterium* spp. including *F. nucleatum* (*n* = 4), *F. varium* (*n* = 3), *F. mortiferum* (*n* = 2), *F. periodonticum* (*n* = 2), and *F. necrophorum* (*n* = 1) presented with low MICs of all antimicrobials tested with few exceptions. Two *F. varium* isolates showed resistance to penicillin, piperacillin/tazobactam, moxifloxacin, and clindamycin and presented with moderate to high MICs of cefoxitin (8 mg/L) and doxycycline (1–2 mg/L) compared to other *Fusobacterium* isolates tested. Additionally, resistance to moxifloxacin was recorded in *F. mortiferum* (2/2) and *F. periodonticum* (1/2).

*Eggerthia catenaformis* demonstrated susceptibility to benzylpenicillin, β-lactam/β-lactamase-inhibitor combinations, and metronidazole. All but one isolate was resistant to moxifloxacin and two isolates were resistant to clindamycin.

When testing *Actinotignum schaalii*, we observed no resistance to β-lactam/β-lactamase-inhibitor combinations and benzylpenicillin. *A. schaalii* presented with resistance to metronidazole which is to be expected. Furthermore, we could observe resistance to moxifloxacin and clindamycin in 11/11 and 5/11 isolates respectively. One isolate presented with a high MIC of doxycycline of >4 mg/L.

The recently renamed *Lancefieldella* spp. including *L. parvula* (*n* = 9), *L. rimae* (*n* = 4) (formerly known as *Atopobium* spp.) showed resistance to clindamycin (5/13), penicillin (2/13), moxifloxacin (1/13), and metronidazole (1/13). Its close relative *Fannyhessae vaginae* (formerly known as *Atopobium vaginae*) showed resistance to metronidazole and a high MIC of doxycycline (4 mg/L).

When comparing the results of the gold standard method (agar dilution method) and the E-tests we observed varying degrees of agreement (Table 3). The essential and categorical agreement was above 90% in ampicillin/sulbactam, meropenem, moxifloxacin, and tigecycline, and 89.2% in imipenem. For metronidazole and clindamycin, the essential agreement was below 90% but the categorical agreement was near or above 90%. Penicillin presented with the lowest categorical agreement of 86.7% and a very high very major error rate of 13.3%. Of the 16 isolates presenting with very major errors, four belonged to *Capnocytophaga* spp. Because cefoxitin and doxycycline have no defined breakpoints, the categorical agreement could not be determined but the essential agreement was 85% and 72.5% respectively. The minimum inhibitory concentration of 50% of the bacteria tested (MIC50) of cefoxitin and doxycycline determined by E-tests was one dilution below the agar dilution and the minimum inhibitory concentration of 90% of the bacteria tested (MIC90) was the same (MIC50: 2/1 mg/L and 0.25/0.125 mg/L; MIC90: 4/4 mg/L and 4/4 mg/L respectively).

## 3. Discussion

Anaerobic bacteria are the most common type of bacteria colonizing our bodies. They can cause severe infections, often in mixed infections with aerobic bacteria. Because of varying resistance rates, it is important to routinely test anaerobic isolates for resistance to commonly used antimicrobials.

This study evaluated the antimicrobial susceptibility of 120 anaerobic isolates belonging to 16 genera to 12 antimicrobial substances. The bacteria tested in this study are rare and not often tested. Therefore, comparing our results to other studies might not be possible. Furthermore, breakpoints are not available for all antibiotics tested and are different depending on the guideline a study follows. In our study, we followed EUCASTs recommendations [8], but CLSI (Clinical and Laboratory Standards Institute) guidelines are used in many studies. This leads to different results, especially in the moxifloxacin resistance rate. Further differences can be explained by the use of various testing methods (e.g., E-tests, broth microdilution, and disc diffusion). Additionally, guidelines for susceptibility testing are available for non-fastidious anaerobes such as *Bacteroides fragilis,* but not for all species tested in this study. The fastidious nature of these rare anaerobes makes them hard to test. The reading of the results was hindered by characteristics such as hazy growth. Almost all isolates could be assessed after only 48 h. Only five isolates (e.g., *Eubacterium callanderi*, *Porphyromonas asaccharolytica*) did not show sufficient growth on the anaerobic growth control plates and therefore could not be evaluated after 48 h. In addition, only a small number of isolates were tested for many species included in this study. This makes tools for the interpretation of susceptibility testing, such as MIC50/90, less reliable and more prone to distortions and makes a numerical comparison of resistance rates impossible in some cases. All these factors have to be taken into consideration when interpreting the results of our study. Because of the uncertainty concerning the validity of the E-tests susceptibility results, the susceptibility data reported in this study stem from agar dilution testing unless stated otherwise, as it is regarded as the gold standard method.

The great susceptibility of all isolates to carbapenems and β-lactam/β-lactamase inhibitor combinations is in line with the literature [9,10,11,12,13]. Penicillin resistance in anaerobes has been reported [11,14,15]. The high rate of moxifloxacin and clindamycin resistance is worrying and seems to be increasing for some of the species tested.

Studies testing *Prevotella* spp. found similar piperacillin/tazobactam, imipenem, meropenem, and clindamycin resistance rates [11,16,17,18,19,20]. Some studies found higher MIC50/90 of doxycycline [17] and similar MIC50/90 of cefoxitin [14]. Studies using the CLSI breakpoint for moxifloxacin find no resistance of *Prevotella* spp. but report a similar MIC50/90 to our study [17]. Studies using EUCAST breakpoints confirm high resistance rates but lower than those observed in our study [11]. The reason for the discrepancies between the agar dilution and E-test results for metronidazole and clindamycin could not be determined and needs further examination.

*Capnocytophaga* spp. are known to produce β-lactamase and therefore often only respond to β-lactam/β-lactamase-inhibitor combinations [9]. This was true for 6/13 isolates tested. High metronidazole and clindamycin MICs have been reported, as well as low MICs of doxycycline [21]. This is in line with our results. The high rate of resistance to metronidazole is to be expected. When testing for penicillin resistance, the agar dilution methodology found a 22.7% resistance while E-tests found only 4.5% resistance.

*Actinotignum schaalii* is intrinsically resistant to metronidazole as a facultative anaerobic bacterium. Cattoir et al. (2010) reported lower rates of resistance to clindamycin and moxifloxacin for *A*. *schaalii* [22], while Hays et al. (2014) found resistance to clindamycin in 21 out of 32 strains tested [23]. The MIC50 and 90 values confirm this is not due to the different breakpoints used. The high susceptibility to β-lactams was confirmed by other studies [12,24,25]. Our finding of a strain with a high MIC of doxycycline is worrying. We could not find another study mentioning this possible resistance. This might be because *A. schaalii*, although being recognized as an emerging uropathogen [12], is not often tested in antimicrobial susceptibility surveillance studies.

*Eggerthia catenaformis* is rarely isolated and has only been documented in case reports. In recent years, more cases of severe infections due to *E. catenaformis* have been reported [26]. Of the 16 isolates tested in this study, 15 were resistant to moxifloxacin, and two to clindamycin. As far as we know, no resistance has been documented in case studies [27,28,29].

*Fusobacterium* spp. presented with resistance to penicillin, moxifloxacin, and clindamycin. These findings can be supported by other surveillance studies [14,15,30]. Often piperacillin/tazobactam is reported as 100% susceptible. Our piperacillin/tazobactam resistance rate of 2/12 isolates is similar to the one reported by Ng, Lily et al. (2015) [30].

The two *Eggerthella lenta* isolates tested showed resistance to penicillin, ampicillin-sulbactam, piperacillin/tazobactam, and moxifloxacin. The MICs of cefoxitin and doxycycline were high, suggesting a possible resistance. Byun et al. (2019) showed resistance to penicillin and moxifloxacin [14]. Other studies found *E. lenta* to be resistant to piperacillin/tazobactam but susceptible to carbapenems and metronidazole which is in line with our results [31,32].

The recently renamed *Lancefieldella* spp. and *Fannyhessae vaginae* (formerly *Atopobium* spp.) are known to be susceptible to piperacillin/tazobactam and carbapenems. Resistance to metronidazole has also been reported [13,17,33]. Some of our isolates tested as resistant to penicillin, moxifloxacin, and clindamycin. To our knowledge, no penicillin resistance has been reported. Badri et al. (2019) tested *Atopobium* spp. using E-tests and reported 100% susceptibility rates to penicillin [13]. This is in line with our results from the E-test method. The lack of resistance reporting might be due to the method used.

*Solobacterium moorei* has been confirmed to be highly susceptible to many antibiotics [34,35]. The isolate tested in this study presented resistance to moxifloxacin, which has not been reported by anyone else.

*Mobiluncus curtsii* showed resistance to metronidazole and clindamycin, as shown in other studies [33,36].

*Alistipes finegoldii* has been shown to produce β-lactamase [37]. Moxifloxacin resistance has been reported for the *Alistipes* genus [38].

*Eubacterium callanderi* has only been reported on in case studies. *E. callanderi* presents with resistance to penicillin, clindamycin, and moxifloxacin [39]. One of the four strains tested resistant to metronidazole, which has not been reported.

*Slackia exigua* presented in our study as highly susceptible to all antibiotics with a breakpoint, except one isolate’s resistance to penicillin. The MICs of doxycycline and cefoxitin were high, but no conclusions on the susceptibility can be drawn due to a lack of breakpoints. Our findings are in line with Kim et al. (2010) who are the only ones to test the susceptibility of *Slackia exigua* to our knowledge [40].

*Porphyromonas asaccharolytica* are generally susceptible to β-lactams as observed in this study [10,17]. Petrina et al. (2019) tested six *Porphyromonas* spp. isolates using agar dilution methodology and found one to be resistant to metronidazole [17]. Of the two isolates tested in our study, one tested as metronidazol resistant using agar dilution but tested as susceptible using E-tests. We cannot explain the discrepancy at this point.

*Trueperella bernardiae* has been shown to be highly susceptible to penicillin, piperacillin/tazobactam, imipenem, and meropenem which is corroborated by our study [41]. We showed resistance to moxifloxacin and metronidazole using both methodologies.

We also compared the agar dilution method to the E-test method for ten antibiotics. Because of the elaborate nature of the agar dilution method, it is not an option for routine susceptibility testing. The bacteria in this study show a wide range of susceptibility profiles and can be resistant to antibiotics routinely used to treat infections involving anaerobic bacteria. Therefore, it is important to evaluate the accuracy of the E-test method. Testing some of the species proved to be rather difficult due to hazy and very slow growth. Because of the limiting factors to our study explained above, our results are only an indication of the accuracy of the E-test method.

While E-tests have been shown to be accurate for all anaerobes in the past [42], other studies have reported problems determining MICs with E-tests of slow-growing bacteria [43]. In our study, the agreement for carbapenems, tigecycline, and ampicillin/sulbactam was over 90% and presented with error rates under 3%. The substances with higher resistance rates often showed high very major error rates, like penicillin (13.3%), or high major error rates like moxifloxacin (7.5%). Because of the lack of breakpoints, cefoxitin and doxycycline could not be fully evaluated. The essential agreement was too low for both of them. Past studies reported that the MICs determined by E-tests are lower than those reported by reference methods [42,44]. We can confirm those findings.

Overall, it seems like penicillin, metronidazole, moxifloxacin, and clindamycin E-tests lead to high error rates. Whether this is due to us using a small number of isolates or due to the fastidious, slow-growing nature of the bacteria has yet to be determined. The impact of slow-growing bacteria, meaning long incubation times has to be investigated further. This could lead to decreasing anti-anaerobic activity of the antibiotics used [43]. Furthermore, we need to establish breakpoints, specifically for anaerobes to allow full evaluation of E-tests for rare anaerobes.

## 4. Materials and Methods

A total of 120 different oral clinical strains collected in previous studies were tested. All bacterial strains were grown on Brucella Agar with 5% defibrinated sheep blood, 1 mg of vit. K/L and 5 mg hemin/L and incubated at 35 ± 2 °C in an anaerobic workstation (Whitley MG 1000, anaerobic workstation, Meintrup Laborgeraete GmbH, Lähden-Holte, Germany) containing 80% N₂, 15% CO₂, and 5% H₂. Bacterial strains were identified using matrix-associated laser desorption ionization time-of-flight mass spectrometry (Vitek-MS system, bioMerieux, Lyon, France).

### 4.1. Agar Dilution

Antibiotics tested were penicillin (benzylpenicillin), ampicillin/sulbactam, piperacillin/tazobactam, imipenem, meropenem, cefoxitin, moxifloxacin, metronidazole, clindamycin, tigecycline, and doxycycline. Due to a lack of availability, eravacycline could not be tested. The ranges of dilution were as follows: penicillin and clindamycin 0.03 mg/L to 2 mg/L; ampicillin/sulbactam 0.5 mg/L to 16 mg/L; piperacillin/tazobactam 0.5 mg/L to 16 mg/L (0.125 mg/L to 4 mg/L for *Prevotella* spp. and *F. necrophorum*); metronidazole 0.5 mg/L to 16 mg/L (0.125 mg/L to 8 mg/L for *Prevotella* spp. and *F. necrophorum*); moxifloxacin 0.06 mg/L to 2 mg/L; tigecycline and doxycycline 0.125 mg/L to 4 mg/L; imipenem and meropenem 0.25 mg/L to 8 mg/L (0.03 mg/L to 1 mg/L for *Prevotella* spp. and *F. necrophorum*); cefoxitin 0.25 mg/L to 8 mg/L (by international convention dilutions below 0.125 mg/L are rounded). For the ampicillin/sulbactam and piperacillin/tazobactam combination, a constant concentration of 4 mg/L of sulbactam or tazobactam was added. Antibiotics were solved in distilled water, acetic acid, or DMSO according to the manufacturer’s instructions. Stock solutions were stored for no longer than six weeks at −20 °C (except imipenem at −80 °C). To ensure the activity of tigecycline was not reduced due to accelerated breakdown in the presence of oxygen [45], 2% Oxyrase was added to the agar or they were stored anaerobically until inoculation.

Geometric dilutions of each antibiotic were added to Brucella Agar with 5% defibrinated sheep blood, 1 mg of vit. K/L and 5 mg hemin/L as recommended by EUCAST. For each dilution two plates were prepared.

The inoculum used was an 0.85% saline solution comparable to 1 McFarland standard. The inoculation was aided with a semi-automatic inoculator device (A400 Multipoint inoculator, Bachofer GmbH, Lähden-Holte, Germany). This produced spots of 5 µL at approximately 10⁵ CFU/spot. Additionally, growth control plates without any antibiotics were inoculated at the beginning and end of the inoculation process and incubated both anaerobically and aerobically. Plates were evaluated 48 h and 96 h after inoculation depending on the colony growth on the growth control plates. MIC was defined as the first concentration that sufficiently inhibits the visible growth of the bacteria.

To verify the accuracy of the results, quality control strains *Eggerthella lenta* ATCC 43,055 and *Bacteroides fragilis* ATCC 25,285 were used during the testing of all bacterial strains. We ensured sterile conditions through incubation of agar plates with no inoculum. Additionally, agar plates with no antibiotics were incubated anaerobically and aerobically to ensure the purity of the inoculum and to ensure adequate colony growth.

### 4.2. E-Tests

Agar plates measuring 150-mm-diameter and containing Brucella agar with 5% defibrinated sheep blood, 1 mg of vit. K/L and 5 mg hemin/L were inoculated with a bacterial suspension in an anaerobic broth comparable to 1 McFarland standard. A maximum of 6 test strips were applied to each plate in a radial fashion at equal distances from one another. Antibiotics tested were: penicillin, ampicillin/sulbactam, imipenem, meropenem, cefoxitin, moxifloxacin, metronidazole, clindamycin, eravacycline, doxycycline, and tigecycline (Liofilchem MTS; bioMerieux, Lyon, France). Due to a lack of availability, piperacillin/tazobactam could not be tested. Plates were read approximately 48 h or 96 h after inoculation depending on the readability of the results. We followed the reading guide provided by the manufacturer.

### 4.3. Statistical Analysis

The MIC-Range, MIC50 (minimum inhibitory concentration of 50% of the bacteria tested), MIC90 (minimum inhibitory concentration of 90% of the bacteria tested), and the rates of susceptibility, intermediate susceptibility, and resistance were determined. The breakpoints used were: penicillin 0.5 mg/L (except *F. necrophorum* 0.06 mg/L), ampicillin/sulbactam 2–8 mg/L, piperacillin/tazobactam 8 mg/L (except *Prevotella* spp. and *F. necrophorum* 0.5 mg/L), imipenem 2–4 mg/L, meropenem 2 mg/L (except *Prevotella* spp. 0.25 mg/L, *F. necrophorum* 0.03 mg/L), clindamycin 0.5 mg/L (except *Prevotella* spp. and *F. necrophorum* 0.25 mg/L), metronidazole 4 mg/L (*F. necrophorum* 0.03 mg/L), moxifloxacin 0.25 mg/L, and tigecycline 0.5 mg/L [8,46].

To compare the gold standard agar dilution to the E-test method we calculated the categorical and essential agreement (CA, EA) and the minor, major, and very major error rates [47]. The essential agreement was defined as the percentage of isolates where the E-test MIC was within ±1 log₂ dilution of the agar dilution MIC. The categorical agreement was defined as the number of isolates where agar dilution and E-test yielded the same result of susceptibility. An antimicrobial susceptibility testing system should have an EA and CA ≥90% and the very major error rate and major error rate should be ≤3%, and the minor error rate ≤10%. The descriptive statistical analysis was performed using IBM SPSS Statistics 28.0.0.0.

## 5. Conclusions

All the strains tested showed high susceptibility to carbapenems, β-lactam/β-lactamase inhibitor combinations, and tigecycline. Trends of increasing resistance to moxifloxacin and clindamycin could be observed for some species. The resistance rates reported in this study are concerning and show the importance of routine susceptibility testing. The low agreement for some antibiotics between the gold standard method and the E-test could be due to the slow growth of the organisms tested. Further investigations are necessary to determine the reason for high error rates and how to improve susceptibility testing of fastidious, rare anaerobes.

## Figures and Tables

**Table 1 antibiotics-12-00063-t001:** Antimicrobial susceptibility of 120 anaerobic bacteria tested with agar dilution method.

		MIC (mg/L)			No. of Isolates with Indicated Susceptibility (%)		
Bacteria	Antibiotic	Range	50%	90%	S	I	R
*Prevotella* spp. *n* = 25	Penicillin	0.03–>2	0.125	>2	16 (64)		9 (36)
	Ampsulb °	0.5	0.5	0.5	25 (100)		
	Piptaz °	0.125–1	0.125	0.125	24 (96)		1 (4)
	Imipenem	0.03–0.06	0.03	0.06	25 (100)		
	Meropenem	0.03–0.125	0.03	0.125	25 (100)		
	Cefoxitin	0.25–4	1	4			
	Metronidazole	0.125–>8	0.5	8	22 (88)		3 (12)
	Moxifloxacin	0.06–>2	1	1	3 (12)		22 (88)
	Clindamycin	0.03–>2	0.5	>2	19 (76)		6 (24)
	Doxycycline	0.125–>4	0.25	4			
	Tigecycline	0.125–0.5	0.125	0.25	25 (100)		
*Capnocytophaga* spp. *n* = 22	Penicillin	0.03–>2	0.125	2	17 (77)		5 (23)
	Ampsulb	0.5–2	0.5	0.5	22 (100)		
	Piptaz	0.5–>16	0.5	0.5	21 (95.5)		1 (4.5)
	Imipenem	0.25–0.5	0.25	0.5	22 (100)		
	Meropenem	0.25–2	0.25	0.5	22 (100)		
	Cefoxitin	1–4	2	2			
	Metronidazole	2–>16	>16	>16	4 (18)		18 (82)
	Moxifloxacin	0.06–>2	0.06	2	19 (86)		3 (14)
	Clindamycin	0.03–>2	>2	>2	7 (32)		15 (68)
	Doxycycline	0.25–1	0.5	1			
	Tigecycline	0.125–0.25	0.25	0.25	22 (100)		
*Fusobacterium* spp. *n* = 12	Penicillin	0.03–1	0.03	1	10 (83)		2 (17)
	Ampsulb	0.5–4	0.5	4	10 (83)	2 (17)	
	Piptaz	0.125–16	0.5	16	10 (83)		2 (17)
	Imipenem	0.03–2	0.25	2	12 (100)		
	Meropenem	0.03–2	0.25	0.5	12 (100)		
	Cefoxitin	0.25–8	4	8			
	Metronidazole	0.25–>16	0.5	0.5	12 (100)		
	Moxifloxacin	0.06–>2	1	>2	6 (50)		6 (50)
	Clindamycin	0.03–>2	0.125	>2	9 (75)		3 (25)
	Doxycycline	0.125–2	0.25	1			
	Tigecycline	0.125–0.25	0.125	0.25	12 (100)		
*Eggerthia catenaformis* *n* = 16	Penicillin	0.03	0.03	0.03	16 (100)		
	Ampsulb	0.5	0.5	0.5	16 (100)		
	Piptaz	0.5	0.5	0.5	16 (100)		
	Imipenem	0.25	0.25	0.25	16 (100)		
	Meropenem	0.25	0.25	0.25	16 (100)		
	Cefoxitin	0.25–2	1	2			
	Metronidazole	0.5–4	2	4	16 (100)		
	Moxifloxacin	0.25–>2	0.5	0.5	1 (6)		15 (94)
	Clindamycin	0.06–>2	0.25	2	14 (87.5)		2 (12.5)
	Doxycycline	0.125–>4	0.125	2			
	Tigecycline	0.125	0.125	0.125	16 (100)		
*Actinotignum schaalii* *n* = 11	Penicillin	0.03–0.5	0.03	0.5	11 (100)		
	Ampsulb	0.5–1	0.5	0.5	11 (100)		
	Piptaz	0.5	0.5	0.5	11 (100)		
	Imipenem	0.25	0.25	0.25	11 (100)		
	Meropenem	0.25	0.25	0.25	11 (100)		
	Cefoxitin	0.25–1	0.25	0.5			
	Metronidazole	>16	>16	>16			11 (100)
	Moxifloxacin	1–2	1	1			11 (100)
	Clindamycin	0.06–>2	>2	>2	5 (45.5)		6 (54.5)
	Doxycycline	0.125–>4	0.25	0.25			
	Tigecycline	0.125–0.5	0.125	0.25	11 (100)		
*Lancefieldella* spp. *n* = 13	Penicillin	0.03–>2	0.125	1	11 (85)		2 (15)
	Ampsulb	0.5	0.5	0.5	13 (100)		
	Piptaz	0.5	0.5	0.5	13 (100)		
	Imipenem	0.25–0.5	0.25	0.5	13 (100)		
	Meropenem	0.25–0.5	0.25	0.25	13 (100)		
	Cefoxitin	1–8	2	4			
	Metronidazole	0.5–>4	1	4	12 (92)		1 (8)
	Moxifloxacin	0.125–>2	0.25	0.25	12 (92)		1 (8)
	Clindamycin	0.03–>2	0.5	>2	8 (61.5)		5 (38.5)
	Doxycycline	0.25–2	1	2			
	Tigecycline	0.125–0.25	0.125	0.125	13 (100)		
*Fannyhessae vaginae* *n* = 1	Penicillin	0.5			1 (100)		
	Ampsulb	0.5			1 (100)		
	Piptaz	0.5			1 (100)		
	Imipenem	0.25			1 (100)		
	Meropenem	0.25			1 (100)		
	Cefoxitin	1					
	Metronidazole	>16					1 (100)
	Moxifloxacin	0.125			1 (100)		
	Clindamycin	0.03			1 (100)		
	Doxycycline	4					
	Tigecycline	0.125			1 (100)		
*Porphyromonas asaccharolytica* *n* = 2	Penicillin	0.03			2 (100)		
	Ampsulb	0.5			2 (100)		
	Piptaz	0.5			2 (100)		
	Imipenem	0.25			2 (100)		
	Meropenem	0.25			2 (100)		
	Cefoxitin	0.03					
	Metronidazole	0.5–8			1 (50)		1 (50)
	Moxifloxacin	0.125			2 (100)		
	Clindamycin	0.03			2 (100)		
	Doxycycline	0.125–0.5					
	Tigecycline	0.125			2 (100)		
*Alistipes finegoldii* *n* = 2	Penicillin	1–>2					2 (100)
	Ampsulb	1			2 (100)		
	Piptaz	0.5–1			2 (100)		
	Imipenem	1–2			2 (100)		
	Meropenem	2			2 (100)		
	Cefoxitin	8					
	Metronidazole	1–>16			1 (50)		1 (50)
	Moxifloxacin	2–>2					2 (100)
	Clindamycin	0.5–2			1 (50)		1 (50)
	Doxycycline	2–4					
	Tigecycline	0.25			2 (100)		
*Odoribacter splanchnicus* *n* = 3	Penicillin	>2					3 (100)
	Ampsulb	0.5			3 (100)		
	Piptaz	0.5			3 (100)		
	Imipenem	0.25–1			3 (100)		
	Meropenem	0.25			3 (100)		
	Cefoxitin	1–2					
	Metronidazole	0.5			3 (100)		
	Moxifloxacin	2–>2					3 (100)
	Clindamycin	0.03			3 (100)		
	Doxycycline	0.125–>4					
	Tigecycline	0.125–0.25			3 (100)		
*Eggerthella lenta * *n* = 2	Penicillin	1–>2					2 (100)
	Ampsulb	2–4				1 (50)	1 (50)
	Piptaz	>16					2 (100)
	Imipenem	2			2 (100)		
	Meropenem	0.5–2			2 (100)		
	Cefoxitin	8–>8					
	Metronidazole	0.5–2			2 (100)		
	Moxifloxacin	0.25–>2			1 (50)		1 (50)
	Clindamycin	0.25–0.5			2 (100)		
	Doxycycline	>4					
	Tigecycline	0.25			2 (100)		
*Eubacterium callanderi* *n* = 4	Penicillin	2–>2					4 (100)
	Ampsulb	0.5			4 (100)		
	Piptaz	0.5			4 (100)		
	Imipenem	0.25			4 (100)		
	Meropenem	0.25			4 (100)		
	Cefoxitin	2					
	Metronidazole	0.5–>16			3 (75)		1 (25)
	Moxifloxacin	1–>2					4 (100)
	Clindamycin	1–>2					4 (100)
	Doxycycline	4–>4					
	Tigecycline	0.125			4 (100)		
*Mobiluncus curtsii* *n* = 1	Penicillin	0.06			1 (100)		
	Ampsulb	0.5			1 (100)		
	Piptaz	4			1 (100)		
	Imipenem	0.25			1 (100)		
	Meropenem	0.25			1 (100)		
	Cefoxitin	2					
	Metronidazole	16					1 (100)
	Moxifloxacin	0.125			1 (100)		
	Clindamycin	>2					1 (100)
	Doxycycline	0.125					
	Tigecycline	0.125			1 (100)		
*Slackia exigua* *n* = 4	Penicillin	0.125–1			3 (75)		1 (25)
	Ampsulb	0.5			4 (100)		
	Piptaz	0.5			4 (100)		
	Imipenem	0.25			4 (100)		
	Meropenem	0.25			4 (100)		
	Cefoxitin	4–8					
	Metronidazole	0.5			4 (100)		
	Moxifloxacin	0.06–0.125			4 (100)		
	Clindamycin	0.03–0.125			4 (100)		
	Doxycycline	0.125–>4					
	Tigecycline	0.125			4 (100)		
*Solobacterium moorei * *n* = 1	Penicillin	0.03			1 (100)		
	Ampsulb	0.5			1 (100)		
	Piptaz	0.5			1 (100)		
	Imipenem	0.25			1 (100)		
	Meropenem	0.25			1 (100)		
	Cefoxitin	1					
	Metronidazole	2			1 (100)		
	Moxifloxacin	0.5					1 (100)
	Clindamycin	0.03			1 (100)		
	Doxycycline	0.25					
	Tigecycline	0.125			1 (100)		
*Trueperella bernardiae* *n* = 1	Penicillin	0.03			1 (100)		
	Ampsulb	0.5			1 (100)		
	Piptaz	0.5			1 (100)		
	Imipenem	0.25			1 (100)		
	Meropenem	0.25			1 (100)		
	Cefoxitin	0.25					
	Metronidazole	>16					1 (100)
	Moxifloxacin	1					1 (100)
	Clindamycin	0.06			1 (100)		
	Doxycycline	0.5					
	Tigecycline	0.125			1 (100)		

° ‘Ampsulb’ stands for ampicillin/sulbactam, ‘piptaz’ stands for piperacillin/tazobactam. By international convention, dilutions below 0.125 are rounded.

**Table 2 antibiotics-12-00063-t002:** Antimicrobial susceptibility of 120 anaerobic bacteria tested with E-test method.

		MIC (mg/L)			No. of Isolates with Indicated Susceptibility (%)		
Bacteria	Antibiotic	Range	50%	90%	S	I	R
*Prevotella* spp. *n* = 25	Penicillin	<0.03–>64	0.125	32	17 (68)		8 (32)
	Ampsulb °	<0.03–1	0.03	0.25	25 (100)		
	Imipenem	<0.03–0.25	0.06	0.125	25 (100)		
	Meropenem	<0.03–0.125	0.06	0.125	24 (96)		1 (4)
	Cefoxitin	<0.03–4	2	4			
	Metronidazole	<0.03–1	0.25	0.5	25 (100)		
	Moxifloxacin	<0.03–>32	1	>32	1 (4)		24 (96)
	Clindamycin	<0.03–>256	<0.03	>256	21 (84)		4 (16)
	Doxycycline	<0.03–16	0.125	8			
	Tigecycline	<0.03–4	0.125	0.25	24 (96)		1 (4)
	Eravacycline	<0.03–0.125	<0.03	0.06			
*Capnocytophaga* spp. *n* = 22	Penicillin	<0.03–2	0.125	20.5	21 (96)		1 (4)
	Ampsulb	<0.03–1	<0.03	0.25	22 (100)		
	Imipenem	<0.03–1	0.25	0.5	22 (100)		
	Meropenem	<0.03–0.25	0.06	0.125	22 (100)		
	Cefoxitin	0.125–8	2	4			
	Metronidazole	1–>256	32	>256	3 (14)		19 (86)
	Moxifloxacin	<0.03–8	<0.03	2	19 (86)		3 (14)
	Clindamycin	<0.03–>256	>256	>256	8 (36)		14 (64)
	Doxycycline	<0.03–0.5	0.125	0.25			
	Tigecycline	<0.03–0.25	0.06	0.125	22 (100)		
	Eravacycline	<0.03–0.03	<0.03	0.03			
*Fusobacterium* spp. *n* = 12	Penicillin	<0.03–0.5	<0.03	0.25	12 (100)		
	Ampsulb	<0.03–1	0.06	1	12 (100)		
	Imipenem	<0.03–2	0.125	1	12 (100)		
	Meropenem	<0.03–0.25	<0.03	0.125	12 (100)		
	Cefoxitin	<0.03–>32	0.25	8			
	Metronidazole	<0.03–>256	0.06	0.125	11 (92)		1 (8)
	Moxifloxacin	0.06–>32	0.5	>32	6 (50)		6 (50)
	Clindamycin	<0.03–>256	0.06	32	8 (67)		4 (33)
	Doxycycline	<0.03–16	0.125	0.5			
	Tigecycline	<0.03–0.5	0.03	0.25	12 (100)		
	Eravacycline	<0.03–0.03	<0.03	<0.03			
*Eggerthia catenaformis* *n* = 16	Penicillin	<0.03–0.06	<0.03	<0.03	16 (100)		
	Ampsulb	<0.03	<0.03	<0.03	16 (100)		
	Imipenem	<0.03–0.03	<0.03	0.03	16 (100)		
	Meropenem	0.03–0.125	0.06	0.125	16 (100)		
	Cefoxitin	<0.03–1	0.5	1			
	Metronidazole	<0.03–4	1	4	16 (100)		
	Moxifloxacin	0.5–1	0.5	1	1 (6)		13 (94)
	Clindamycin	0.03–>256	0.25	>256	13 (81)		3 (19)
	Doxycycline	<0.03–32	0.1250.03	16			
	Tigecycline	<0.03–0.03	<0.03	<0.03	16 (100)		
	Eravacycline	<0.03	<0.03	<0.03			
*Actinotignum schaalii* *n* = 11	Penicillin	<0.03–0.06	<0.03	0.03	11 (100)		
	Ampsulb	<0.03–0.03	<0.03	<0.03	11 (100)		
	Imipenem	<0.03–0.25	<0.03	<0.03	11 (100)		
	Meropenem	<0.03–0.25	<0.03	0.06	11 (100)		
	Cefoxitin	0.25–1	0.25	0.5			
	Metronidazole	>256	>256	>256			11 (100)
	Moxifloxacin	0.5–8	1	4			11 (100)
	Clindamycin	<0.03–>256	0.125	>256	7 (63.6)		4 (36.4)
	Doxycycline	<0.03–8	0.06	0.125			
	Tigecycline	<0.03–0.25	<0.03	0.25	11 (100)		
	Eravacycline	<0.03–0.06	<0.03	0.03			
*Lancefieldella* spp. *n* = 13	Penicillin	<0.03–0.25	0.06	0.25	13 (100)		
	Ampsulb	<0.03–0.125	<0.03	0.06	13 (100)		
	Imipenem	<0.03–0.125	<0.03	0.125	13 (100)		
	Meropenem	<0.03–0.5	0.125	0.25	13 (100)		
	Cefoxitin	0.5–8	2	4			
	Metronidazole	0.06–>256	0.5	>256	12 (92)		1 (8)
	Moxifloxacin	0.06–>32	0.25	0.5	9 (71)		4 (29)
	Clindamycin	<0.03–8	1	4	6 (46)		7 (54)
	Doxycycline	<0.03–1	0.5	1			
	Tigecycline	<0.03–0.25	0.125	0.125	13 (100)		
	Eravacycline	<0.03–0.06	0.03	0.06			
*Fannyhessae vaginae* *n* = 1	Penicillin	0.125			1 (100)		
	Ampsulb	0.06			1 (100)		
	Imipenem	<0.03			1 (100)		
	Meropenem	<0.03			1 (100)		
	Cefoxitin	2					
	Metronidazole	>256					1 (100)
	Moxifloxacin	0.125			1 (100)		
	Clindamycin	<0.03			1 (100)		
	Doxycycline	4					
	Tigecycline	0.03			1 (100)		
	Eravacycline	<0.03					
*Porphyromonas asaccharolytica* *n* = 2	Penicillin	<0.03–0.06			2 (100)		
	Ampsulb	<0.03			2 (100)		
	Imipenem	<0.03–0.06			2 (100)		
	Meropenem	<0.03			2 (100)		
	Cefoxitin	0.125–0.5					
	Metronidazole	<0.03			2 (100)		
	Moxifloxacin	0.125			2 (100)		
	Clindamycin	<0.03–64			2 (100)		
	Doxycycline	<0.03–2					
	Tigecycline	<0.03–0.06			2 (100)		
	Eravacycline	<0.03					
*Alistipes finegoldii* *n* = 2	Penicillin	<0.03–2			1 (50)		1 (50)
	Ampsulb	<0.03–0.125			2 (100)		
	Imipenem	<0.03–1			2 (100)		
	Meropenem	<0.03–0.25			2 (100)		
	Cefoxitin	0.06–4					
	Metronidazole	<0.03–0.125			2 (100)		
	Moxifloxacin	0.5–>32					2 (100)
	Clindamycin	0.25–2			1 (50)		1 (50)
	Doxycycline	0.125–2					
	Tigecycline	0.06–0.25			2 (100)		
	Eravacycline	<0.03					
*Odoribacter splanchnicus* *n* = 3	Penicillin	<0.03–2			2 (67)		1 (33)
	Ampsulb	<0.03			3 (100)		
	Imipenem	<0.03–0.03			3 (100)		
	Meropenem	<0.03			3 (100)		
	Cefoxitin	<0.03–1					
	Metronidazole	<0.03			3 (100)		
	Moxifloxacin	2–>32					3 (100)
	Clindamycin	<0.03			3 (100)		
	Doxycycline	<0.03–4					
	Tigecycline	<0.03–0.03			3 (100)		
	Eravacycline	<0.03					
*Eggerthella lenta * *n* = 2	Penicillin	1–4					2 (100)
	Ampsulb	0.125–2			2 (100)		
	Imipenem	0.125–2			2 (100)		
	Meropenem	0.5			2 (100)		
	Cefoxitin	8					
	Metronidazole	0.125			2 (100)		
	Moxifloxacin	0.25–4			1 (50)		1 (50)
	Clindamycin	0.125–0.5			2 (100)		
	Doxycycline	8					
	Tigecycline	0.06			2 (100)		
	Eravacycline	<0.03–0.03					
*Eubacterium callanderi* *n* = 4	Penicillin	0.25–2			1 (25)		3 (75)
	Ampsulb	0.06–0.125			4 (100)		
	Imipenem	<0.03–0.03			4 (100)		
	Meropenem	<0.03–0.06			4 (100)		
	Cefoxitin	1–2					
	Metronidazole	<0.03–>256			1 (25)		3 (75)
	Moxifloxacin	0.25–>32					4 (100)
	Clindamycin	0.5–>256					4 (100)
	Doxycycline	1–4					
	Tigecycline	<0.03–0.06			4 (100)		
	Eravacycline	<0.03–0.03					
*Mobiluncus curtsii* *n* = 1	Penicillin	<0.03			1 (100)		
	Ampsulb	0.06			1 (100)		
	Imipenem	0.06			1 (100)		
	Meropenem	0.03			1 (100)		
	Cefoxitin	2					
	Metronidazole	32					1 (100)
	Moxifloxacin	0.03			1 (100)		
	Clindamycin	>256					1 (100)
	Doxycycline	0.125					
	Tigecycline	0.03			1 (100)		
	Eravacycline	0.03					
*Slackia exigua* *n* = 4	Penicillin	<0.03–0.25			4 (100)		
	Ampsulb	<0.03–0.25			4 (100)		
	Imipenem	<0.03–0.06			4 (100)		
	Meropenem	0.03–0.06			4 (100)		
	Cefoxitin	4–8					
	Metronidazole	<0.03–0.06			4 (100)		
	Moxifloxacin	0.06			4 (100)		
	Clindamycin	<0.03			4 (100)		
	Doxycycline	0.06–24					
	Tigecycline	<0.03–0.125			4 (100)		
	Eravacycline	<0.03					
*Solobacterium moorei * *n* = 1	Penicillin	<0.03			1 (100)		
	Ampsulb	<0.03			1 (100)		
	Imipenem	<0.03			1 (100)		
	Meropenem	0.06			1 (100)		
	Cefoxitin	0.25					
	Metronidazole	0.25			1 (100)		
	Moxifloxacin	0.5					1 (100)
	Clindamycin	0.125			1 (100)		
	Doxycycline	0.125					
	Tigecycline	0.125			1 (100)		
	Eravacycline	<0.03					
*Trueperella bernardiae* *n* = 1	Penicillin	<0.03			1 (100)		
	Ampsulb	0.03			1 (100)		
	Imipenem	0.03			1 (100)		
	Meropenem	0.03			1 (100)		
	Cefoxitin	0.06					
	Metronidazole	>256					1 (100)
	Moxifloxacin	1					1 (100)
	Clindamycin	0.06			1 (100)		
	Doxycycline	0.25					
	Tigecycline	0.06			1 (100)		
	Eravacycline	<0.03					

° ‘Ampsulb’ stands for ampicillin/sulbactam, ‘piptaz’ stands for piperacillin/tazobactam. By international convention dilutions below 0.125 are rounded.

**Table 3 antibiotics-12-00063-t003:** Comparison of the results obtained by agar dilution method and by E-Test method.

Antibiotic	Essential Agreement (%)	Categorical Agreement (%)	Minor Error (%)	Major Error (%)	Very Major Error (%)
Penicillin	90/120 (75)	104/120 (86.7)			16/120 (13.3)
Ampicillin/sulbactam	116/120 (96.7)	117/120 (97.5)	3/120 (2.5)		
Imipenem	107/120 (89.2)	120/120 (100)			
Meropenem	112/120 (93.3)	119/120 (99.2)		1/120 (0.8)	
Cefoxitin	102/120 (85)				
Metronidazole *	62/87 (71.3)	78/87 (89.7)		3/87 (3.4)	6/87 (6.9)
Moxifloxacin	108/120 (90)	109/120 (91)		8/120 (7.5)	3/120 (2.5)
Clindamycin	99/120 (82.5)	109/120 (90.8)		5/120 (4.5)	6/120 (5)
Doxycycline	87/120 (72.5)				
Tigecycline	116/120 (96.7)	119/120 (99.2)		1/120 (0.8)	

* Metronidazole: all isolates except intrinsically resistant *Actinotignum schaalii* (*n* = 11) and *Capnocytophaga* spp. (*n* = 22).

## Data Availability

The data presented in this article is available in Table 1, Table 2 and Table 3.

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
