# Peer review of "Antimicrobial Susceptibility Profile of Rare Anaerobic Bacteria"

_antibiotics, 2022, doi:10.3390/antibiotics12010063_

Round 1

Reviewer 1 Report

The authors performed antimicrobial susceptibility test on rare anaerobic bacteria against 12 antimicrobials using agar dilution method and E-Test method. Results showed low resistance to β-lactam/β-lactamase inhibitor combinations and no resistance to carbapenems and tigecycline, while resistance to moxifloxacin and clindamycin were observed. The comparison of the two methods was also performed. This work provided resistance data on these rare species which could be important reference for clinicians and researchers. This manuscript is well organized and well written. There are only a few comments/suggestions:

1.     The authors claimed that 120 rare anaerobes were tested. While 120 refers to the total number of isolates, it is misleading to the readers that there were 120 different anaerobe species tested. Please clarify.

2.     Line 102-10: “The essential and categorical agreement was over 90% in ampicillin/sulbactam, imipenem, meropenem, moxifloxacin, and tigecycline.” According to Table 3, imipenem’s EA is 89.2%.

Author Response

We thank reviewer 1 for their helpful comments. The changes based on their feedback are highlighted in red.

  1. The authors claimed that 120 rare anaerobes were tested. While 120 refers to the total number of isolates, it is misleading to the readers that there were 120 different anaerobe species tested. Please clarify.

We clarified this in the manuscript in lines 7 and 125-126 (revised text: 120 rare anaerobic isolates belonging to 16 genera)

  1. Line 102-10: “The essential and categorical agreement was over 90% in ampicillin/sulbactam, imipenem, meropenem, moxifloxacin, and tigecycline.” According to Table 3, imipenem’s EA is 89.2%.

We made changes to the manuscript in lines 15 (in the revised text we deleted imipenem), 104-105 (revised text: The essential and categorical agreement was above 90 % in ampicillin/sulbactam, meropenem, moxifloxacin, and tigecycline, and 89,2% in imipenem).  

Reviewer 2 Report

1. In lines 51-51, for the antimicrobial substances with no EUCAST breakpoints (cefoxitin, doxycycline, and eravacycline) no susceptibility rates could be calculated. As far as I know, at least CLSI has the breakpoints of cefoxitin. In addition to EUCASTCLSI and FDA also develop breakpoints for anaerobic AST (PMID: 24867792). There are differences in interpretive criteria between the organizations, such as the differences in dosages, administration intervals, inoculum size, or test media. Do the authors compare the differences between EUCAST and CLSI on the procedural guidelines for anaerobic AST? And why do the authors select EUCAST breakpoints to interpret the antimicrobial susceptibility testing results ?
2. Is the essential agreement and categorical agreement used by other studies? Please illustrate.
3. In line 244, AST plates were incubated at 37 °C. Maybe 35 ± 2ºC in anaerobic conditions is the optimum culture condition (PMID: 34520862)
4. In line 272, is it multipoint inoculator A400 (Denley Ltd, Billingshurst, Sussex)? It should be written detailed. The inoculating needle is made of stainless steel, is it ? How to ensure sterility before inoculating bacteria, please explain. Which dose to inoculate to one plate each time, 0.3μl1μl2μl5μl ? And the final inoculum is how many CFU per spot. Every plate should have a spot free of growth (negative control).
5. In line 274, Plates were evaluated 48 hours and 96 hours after inoculation. However, other papers just need 48 hours of incubation (PMID: 24867792) (PMID: 34520862). I am too worried that some antibiotics will inactivate at the temperature of 35 ± 2ºC for 96 hours.
6. I suggest a Quality Control section in the Materials and Methods, including before, middle and after the agar dilution test. How to read the results about the hazy growth, confluent colonies, several normal-sized colonies? 

Author Response

We thank reviewer 2 for their thorough feedback. The changes based on their comments are highlighted in yellow.

  1. In lines 51-51, for the antimicrobial substances with no EUCAST breakpoints (cefoxitin, doxycycline, and eravacycline) no susceptibility rates could be calculated. As far as I know, at least CLSI has the breakpoints of cefoxitin. In addition to EUCAST, CLSI and FDA also develop breakpoints for anaerobic AST (PMID: 24867792 ). There are differences in interpretive criteria between the organizations, such as the differences in dosages, administration intervals, inoculum size, or test media. Do the authors compare the differences between EUCAST and CLSI on the procedural guidelines for anaerobic AST? And why do the authors select EUCAST breakpoints to interpret the antimicrobial susceptibility testing results ?

We decided to solely use the EUCAST breakpoints and methodology guidelines because of the open access and the fact that it’s a European Organization. EUCAST documents are regularly used in our university clinic and affiliated laboratories. The aim of our study was not to compare the guidelines and methodologies of the different organizations. Instead, we wanted to provide information on susceptibility rates of rare anaerobic bacteria, using the methodology accessible to our university and laboratory in our country and Europe.

  1. Is the essential agreement and categorical agreement used by other studies? Please illustrate.

The essential and categorical agreement is used when comparing AST methods. The studies cited in the discussion used these values but did not call them CA/EA. They were referred to as “no. of discrepancies/ Percentage of isolates for which MICs differed by the following numbers of log dilutions” [doi: 10.1093/jac/35.6.775] or “ No. of categorical discrepancies / ± 1 dilution % within” [doi: 10.1128/jcm.29.10.2197-2203.1991]. We followed the guideline ISO 20776-1, which defines EA and CA. Additionally, this guidance document defines the rates to classify an alternative testing method as accurate or inaccurate which we listed in lines 312-314. We have added the reference to this document in the methods section in line 309 (new reference number: [49]).

Here are two studies that used EA/CA when comparing AST methods:

Agreement of Quantitative and Qualitative Antimicrobial Susceptibility Testing Methodologies: The Case of Enrofloxacin and Avian Pathogenic Escherichia coli (doi: 10.3389/fmicb.2020.570975)

Antimicrobial susceptibility testing of colistin – evaluation of seven commercial MIC products against standard broth microdilution for Escherichia coli, Klebsiella pneumoniae, Pseudomonas aeruginosa, and Acinetobacter spp. (https://doi.org/10.1016/j.cmi.2017.11.020)

  1. In line 244, AST plates were incubated at 37 °C. Maybe 35 ± 2ºC in anaerobic conditions is the optimum culture condition (PMID: 34520862 )

The temperature readings of our anaerobic workstation varied between 35 and 37 degrees Celsius. The change has been added to the manuscript in line 249 (revised text: 35 ± 2ºC ).

  1. In line 272, is it multipoint inoculator A400 (Denley Ltd, Billingshurst, Sussex)? It should be written detailed. The inoculating needle is made of stainless steel, is it ? How to ensure sterility before inoculating bacteria, please explain. Which dose to inoculate to one plate each time, 0.3μl、1μl、2μl、5μl ? And the final inoculum is how many CFU per spot. Every plate should have a spot free of growth (negative control).

The multipoint inoculator is produced by Bachofer GmbH, Germany. The needles are made of stainless steel and were disinfected and sterilized according to the routine laboratory procedure. This included 24h soak in disinfectant and a sterilization process (autoclaving). The inoculation process resulted in approximately 10^5 CFU/spot (5 microliters/spot). The negative controls were done at the beginning and end of the inoculation process. We edited the section you are referring to in lines 276-277 (revised text: The inoculation was aided with a semi-automatic inoculator device (A400 Multipoint inoculator, Bachofer GmbH, Germany). This produced spots of 5 µL at approximately 10⁵ CFU/spot.)

  1. In line 274, Plates were evaluated 48 hours and 96 hours after inoculation. However, other papers just need 48 hours of incubation (PMID: 24867792 ) (PMID: 34520862 ). I am too worried that some antibiotics will inactivate at the temperature of 35 ± 2ºC for 96 hours.

Almost all isolates could be assessed after only 48h. Rarely isolates (for example Eubacterium callanderi, Porphyromonas asaccharolytica) did not show sufficient growth on the anaerobic growth control plates and therefore could not be evaluated after 48h. This presents a problem with the AST of certain slow-growing bacteria. While we did point out the problem of the inactivation of antibiotics in the discussion (238-240), we have clarified the incubation time in lines 137-149 (revised text: Almost all isolates could be assessed after only 48h. Only five isolates (e.g. Eubacterium callanderi, Porphyromonas asaccharolytica) did not show sufficient growth on the anaerobic growth control plates and therefore could not be evaluated after 48h.)

  1. I suggest a Quality Control section in the Materials and Methods, including before, middle and after the agar dilution test. How to read the results about the hazy growth, confluent colonies, several normal-sized colonies? 

We added comments on quality control to 4.1 agar dilution in lines 284-286 (revised text: We ensured sterile conditions through incubation of agar plates with no inoculum. Additionally, agar plates with no antibiotics were incubated anaerobically and aerobically to ensure the purity of the inoculum and to ensure adequate colony growth.).

During agar dilution testing, the difference between hazy growth and normal colonies was determined via the elevation of the colonies. The determination was aided by comparing growth on the growth control plates with the spot in question. We defined the MIC as the first concentration that visibly inhibits the growth of the bacteria. When reading the E-Tests we followed the reading instructions provided by the manufacturer. We clarified this in lines 296-297 (revised text: We followed the reading guide provided by the manufacturer.)